# Improving Bionic Limb Control through Reinforcement Learning in an Interactive Game Environment

**Kilian Freitag** [1 2]  **Rita Laezza** [1]  **Jan Zbinden** [1 2]  **Max Ortiz-Catalan** [1 2 3]

## Abstract

Enhancing the accuracy and robustness of bionic limb controllers that decode motor intent is a pressing challenge in the field of prosthetics. State-of-the-art research has mostly focused on Supervised Learning techniques to tackle this problem. However, obtaining high-quality labeled data that accurately represents muscle activity during daily usage remains difficult. In this work, we investigate the potential of Reinforcement Learning to further improve the decoding of human motion intent by incorporating usage-based data. We propose a new method which starts with a control policy, pretrained on a static recording of electromyograhic (EMG) ground truth data. We then fine-tune the pretrained classifier with dynamic EMG data obtained during interaction with a game environment developed for this work. We evaluate our approach in real-time experiments, showing substantial improvements for human-in-the-loop performance. The method proves more effective in predicting simultaneous finger movements, doubling the decoding accuracy both during gameplay and in a separate motion test. See our project page for visual demonstrations sites.google.com/view/bionic-limb-rl.

## 1. Introduction

Roughly 58 million people were living with limb amputation worldwide, as of 2017 (McDonald et al., 2021). Even though extensive research efforts have gone into developing prosthetic devices, these are still far from human-level performance. As a consequence, the adoption of prostheses is effortful, and abandonment rates are high, with one of the main factors being a lack of reliable functionality (Cordella et al., 2016; Smail et al., 2021). The performance of a prosthesis greatly depends on the accuracy with which the user's motion intent can be decoded to determine the appropriate bionic joint actuation. There are several biological signals that can be used to determine human motion intent, such as electroencephalography (EEG) or electromyography (EMG) measurements. The latter has been shown to be more practical and is more widely adopted (Jiang et al., 2010; Li et al., 2021), and is therefore used in this work. EMG can be recorded either through surface or implanted electrodes (Ortiz-Catalan et al., 2014). In myoelectric prosthetic control, there are two main approaches used to decode motor intent for actuating bionic limb joints:

**Direct control**, involves mapping individual muscle signals, often on a one-to-one basis, to a specific bionic joint. When a mapped muscle contracts and its signal value surpasses a predetermined threshold, the corresponding bionic joint is actuated. This approach is appealing due to its intuitiveness and simplicity, but its applicability is limited because many muscles, and thereby their signals, are lost during amputation. Consequently, people with amputation are often left with a sparse set of usable signals. Moreover, the few signals available are often not easily separable and thus cannot always be mapped in a one-to-one fashion. In practice, direct control frequently involves mapping only two antagonistic muscle groups, with mode switching employed to control more than one Degree of Freedom (DOF). For example, the user might sustain an open hand movement to switch between types of hand grasps.

**Pattern recognition control**, employs Machine Learning (ML) algorithms to train a function approximator. This method addresses the issue of signal separability, allowing the user's biological signals to be mapped to the intended motion. In other words, the user can execute different contraction patterns, and the ML algorithms decode the associated motor intent (Kuiken et al., 2016). According to Mereu et al. (2021), this approach is preferred over direct control by people with amputation because it is more intuitive. Many ML approaches have been tested to reliably decode motor intent, from classical algorithms like support

---

[1]Division of Systems and Control, Department of Electrical Engineering, Chalmers University of Technology, Gothenburg, Sweden [2]Center for Bionics and Pain Research, Mölndal, Sweden [3]Bionics Institute, Melbourne, Australia. Correspondence to: Kilian Freitag <tamino@chalmers.se>.

*Interactive Learning with Implicit Human Feedback Workshop at the $40^{th}$ International Conference on Machine Learning*, Honolulu, Hawaii, USA. PMLR 202, 2023. Copyright 2023 by the author(s).

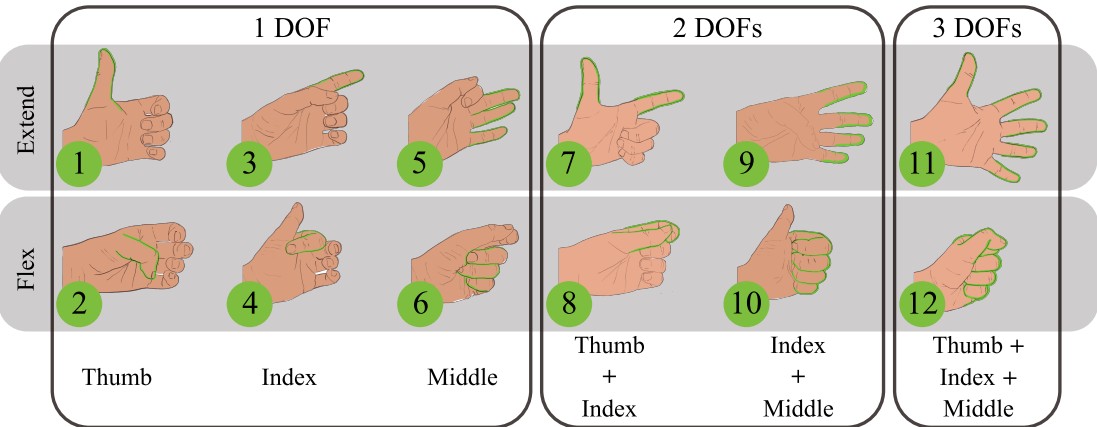

*Figure 1.* Selected finger movements, grouped by number of simultaneous DOFs. Top row consists of finger extension movements, while bottom row consists of finger flexion movements. Each movement is labeled as $m_i$ with $i = 0, \ldots, 12$ and with $m_0$ referring to 'Rest'.

vector machines (Oskoei & Hu, 2008), to deep learning algorithms using Artificial Neural Networks (ANNs). Previous work on pattern recognition control has primarily focused on Supervised Learning (SL) approaches with different ANN architectures, such as feed-forward (Hudgins et al., 1993), recurrent (Williams et al., 2022; Luu et al., 2022) and transformers (Godoy et al., 2022).

One of the drawbacks of SL methods is that they require labeled EMG data. This typically involves instructing the user to repeatedly contract their muscles to generate activation patterns for each movement that is to be trained during a lengthy recording session. Collecting such ground-truth data becomes increasingly cumbersome as the number of DOFs grows. The problem is only exacerbated when simultaneous movements are also considered (e.g. open hand while rotating wrist) since each movement combination needs to be present in the data.

In this work, we propose a novel training procedure for simultaneous myoelectric control based on Reinforcement Learning (RL) to **(a)** record data closely resembling real-life scenarios **(b)** increase viable training tasks and **(c)** help mitigate potentially cumbersome recordings. In contrast to SL methods, RL allows learning by trial-and-error while interacting with an environment. Moreover, instead of trying to clone the behavior observed in the recording session, RL can optimize a policy based on interactive data and a reward function. By selecting a reward function which encodes accurate decoding of motor intent, we aim to improve the performance of myoelectric control policies. Notably, RL can be applied to any Markovian task that gives a measure of success, opening up the possibility for users to give direct feedback to the control policy, based on their preferences.

To that end, we developed a simplified training environment, which allows for a quantifiable measure of improvement. The training environment includes a game inspired by Guitar

Hero, which requires the execution of the correct motions at the correct time and for the appropriate duration. The task was developed especially with a simultaneous control setting in mind since any combination of movements can be implemented. Our results demonstrate the efficacy of RL in improving the decoding of motor intent, with a three-fold increase in normalized cumulative reward (from 0.26 to 0.78) and a more than two-fold increase in Exact Match Ratio (EMR) compared to a policy pretrained using SL. Additionally, the generalizability of our approach is validated through testing on a separate task, also revealing a two-fold increase in EMR and thus reinforcing our findings.

## 2. Related Work

RL methods for prosthetics have received limited exploration. Nevertheless, some promising approaches have emerged in the literature. In this section, we highlight some of the related work on RL for myoelectric control.

For lower limb prosthetics, De Vree & Carloni (2021) showed that simulation-based RL can help to study gate patterns. However, erratic patterns observed in simulations limit their direct applicability as control mechanisms in real-life scenarios. Wen et al. (2019) propose an RL-based method that uses position, velocity, and force measurements from bionic legs as inputs, allowing for personalized knee control. This approach enables online learning of parameters and has been tested on both able-bodied individuals and amputees. Wu et al. (2022) builds upon this work by additionally tracking the intact knee for improved control.

For upper limb prosthetics, Edwards et al. (2016) present an RL-based approach that learns adaptive mode switching. This method enhances the usability of upper limb direct prosthetic control by reducing the number of mode switches required to perform tasks. Pilarski et al. (2011) propose an

RL method that predicts arm movements using EMG and robot state information. The agent learns to match the human arm angle using velocity-based control. They initially guide the learning through a reward based on proximity to the desired angle and, subsequently through sparse human feedback. Vasan & Pilarski (2017) presented a system that allows for the control of 3 simultaneous DOFs, of a prosthetic limb. This was achieved by employing RL, to train an ANN while recording EMG data of hand movements. Their method was able to predict proportionality for each DOF and was successfully tested on three able-bodied subjects.

## 3. Motivation

This work is motivated by limiting factors, which prevent the development of intuitive and reliable control of prostheses. There are three main aspects which we want to highlight:

Firstly, the real-time usability of ML-based controllers remains limited, especially for hand gestures (Li et al., 2021). Indeed, much of prosthetics research using ML methods is confined to evaluations using pre-recorded, offline data. This is notable, as high offline performance in decoding motor intent does not guarantee the same online performance, with a human in the loop (Ortiz-Catalan et al., 2015). As we aim to reach ML solutions which are clinically relevant and of practical applicability in real-life settings, we prioritize low runtime complexity and focus on improvements in online, human-in-the-loop tests.

Secondly, a majority of ML research in prosthetics relies on SL methods using data from recording sessions, which can differ from natural muscle activity during daily usage. This discrepancy between training data and real-life usage can hinder robust control and thereby impact the functionality of ML-based prostheses. To address this issue, we propose training a classifier within a more interactive setting, which can be more representative of daily life scenarios. In essence, we choose to apply RL on interactive data, in order to bridge the distributional gap between training and usage. Moreover, by having the user interact with the environment with the learned RL policy, there is an implicit human feedback embedded in the collected data.

Thirdly, as discussed in Vasan & Pilarski (2017), by formulating the prosthetic control task as an RL problem we also open up the possibility to apply our method to a wider range of tasks. While in this work we present a proof-of-concept for a specific game environment, one could adapt our methods to include real-life tasks, tailored to the patient's need. Specifically, it could allow for personalized adaptation through direct user feedback, which has been demonstrated to be a powerful approach, referred to as Reinforcement Learning from Human Feedback (RLHF) (Ouyang et al., 2022).

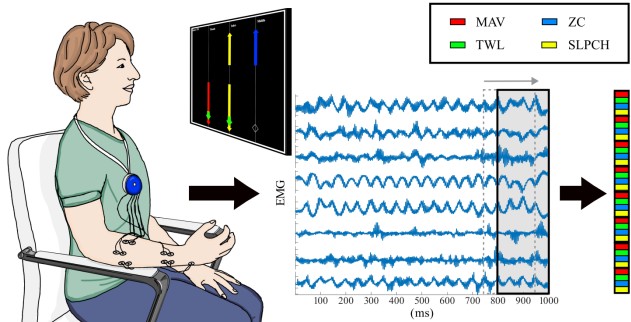

*Figure 2.* EMG recording setup, with sliding window over 8 input channels. The Hudgins et al. (1993) features are extracted for each recorded channel.

## 4. Preliminaries

In this section we aim to introduce the key concepts which make up our proposed method, namely myoelectric control in Section 4.1 and RL in Section 4.2.

### 4.1. Myoelectric Control

When designing a myoelectric controller, a key choice is to decide on the specific movements to control. This decision hinges on the participant's amputation level; greater proximal limb loss precludes control of movements tied to now-absent muscles. Although our method is broadly applicable, we focus on finger movements corresponding to common grasp patterns, shown in Fig. 1. Incongruent articulations (flex + extend combinations) are not selected as there is evidence that natural control of such motions is not simultaneous (Rosenbaum, 2009). This selection of finger movements would be feasible for people with trans-radial amputation. Alternatively, they would be reasonable for patients who have lost the limb more proximal but underwent nerve transfer surgery to create additional myoelectric sites (Ortiz-Catalan et al., 2020; Osborn et al., 2021).

To acquire EMG signals, eight surface electrodes are placed on the forearm of the participant, with an additional electrode for ground, as illustrated in Fig. 2. The electrodes are placed in a bipolar configuration, i.e. for each channel, two adjacent signals are subtracted from one another to reduce noise. The EMG data stream of these eight channels is split with a sliding window approach, into 200 ms windows with 150 ms overlap, equating to an update frequency of 20 Hz.

A common approach to overcome the problem of having high dimensional raw EMG data is to extract a set of four features proposed by Hudgins et al. (1993) from the windowed EMG data. The features in question are: mean absolute value (MAV), waveform length in time-domain (TWL), number of zero crossings (ZC) and slope changes (SLPCH). The input to the myoelectric control policy is the stacked vector of these features for each channel (see Fig. 2).

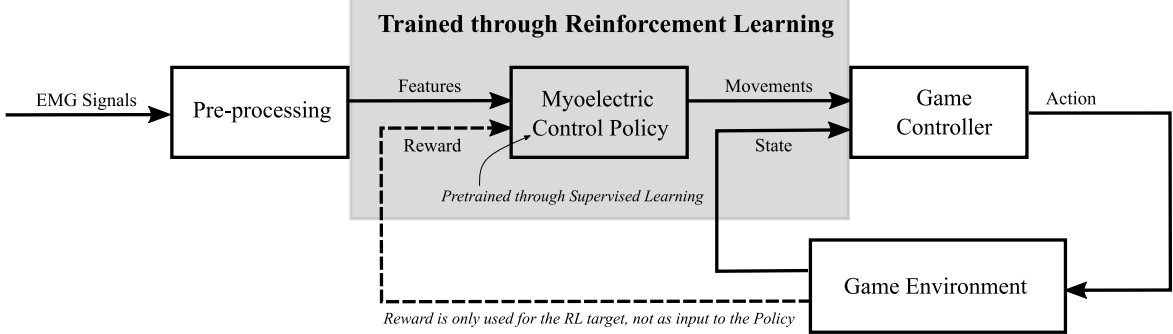

*Figure 3.* The used RL framework consists of obtaining EMG signals from humans, that are given to the policy to perform actions in an environment. This environment then gives a reward based on how successful an action was.

Note how for each DOF (Thumb, Index and Middle, see Fig. 1), there are two movements (Flex and Extend). Clinically, sequential myoelectric control is prevalent. This means that for each possible DOF that a prosthesis can actuate, only one can be active at any given time ($m_1$ to $m_6$). In such cases, the ML problem is formulated as a simple classification task. However, since dexterous manipulation requires simultaneous actions, there have been attempts to control multiple DOFs at the same time (Jiang et al., 2008; Ameri et al., 2018). The most straightforward approach is to treat each movement combination as a new class. Alternatively, we can formulate the ML problem as a multi-label classification task instead. To that end, each movement is encoded into a binary vector $m_i \in \{0, 1\}^{2 \cdot \text{DOF}+1}$, with DOF = 3. For example, simultaneous thumb and index flexion is encoded as:

$$m_8 = \begin{bmatrix} \underbrace{0 \quad 1}_{\text{Thumb}} & \underbrace{0 \quad 1}_{\text{Index}} & \underbrace{0 \quad 0}_{\text{Middle}} & \underbrace{0}_{\text{Rest}} \end{bmatrix} \quad (1)$$

The control policy is chosen to be a relatively small feed-forward, fully-connected (6 hidden layers with ReLU activation, each with 128 neurons) ANN architecture (see Molin (2022) for more details), in order to allow easy transfer to an embedded device. The output layer has a sigmoid activation function, outputting values between $[0, 1]$. At test time, outputs are rounded to be exactly $\{0, 1\}$.

To evaluate the policy's capability of decoding motor intent, we calculate the EMR and the F1 macro scores. In the context of ML, the EMR corresponds to the classification accuracy, however, in a multi-label setting, this terminology can be ambiguous. For this reason, we refer to the EMR instead, which measures the proportion of correct predictions out of all predictions made by the classifier. When computing the EMR, even partially correct answers are considered completely incorrect, emphasizing the requirement for precise and accurate predictions. Nevertheless, if some but not all of the target labels are predicted, one could argue that this is more accurate than a case in which none of the target labels are predicted. Therefore we additionally consider the

F1 score for evaluation, which offers a class-wise assessment of performance and is a commonly used indicator for multi-label classification tasks (Yang, 1999). The F1 score for each class, $i = 1, \ldots, N$, is computed as:

$$\text{F1}_i = \frac{\text{TP}_i}{\text{TP}_i + 0.5(\text{FP}_i + \text{FN}_i)} \quad (2)$$

where TP denotes True Positives, FP represents False Positives, and FN corresponds to False Negatives. The F1 macro score is obtained by taking the macro average:

$$\text{F1}_{\text{macro}} = \frac{1}{N} \sum_{i=1}^{N} \text{F1}_i \quad (3)$$

Together, these evaluation metrics provide comprehensive insights into the effectiveness of the classifier for multi-label classification tasks.

### 4.2. Reinforcement Learning

The idea behind RL is to learn a policy from trial-and-error while interacting with the environment which provides a reward signal. More formally, RL problems are formulated as Markov Decision Processes (MDPs). In this work, we consider an episodic MDP setting, defined as a tuple $(\mathcal{S}, \mathcal{A}, p, r, \gamma)$, where $r : \mathcal{S} \to \mathbb{R}$ is a reward function and $\gamma \in (0, 1]$ is the discount factor. $\mathcal{S}$ and $\mathcal{A}$ are the state and action spaces, respectively. The probability density function $p(s_{t+1}|s_t, a_t)$ represents the probability of transitioning to state $s_{t+1}$, given the current state $s_t$ and action $a_t$, with $s_t, s_{t+1} \in \mathcal{S}$ and $a_t \in \mathcal{A}$.

We aim to learn an ANN policy $\pi_\theta(s_t) = a_t$, i.e. the actor, parameterized by $\theta$. The long-term objective function is defined by the return, which is the sum of discounted future rewards: $G_t = \sum_{k=t}^{T} \gamma^{k-t} r(s_k)$, where $t$ and $T$ are the current and terminal state's indices, respectively. RL algorithms aim to maximize the expected return conditioned on states, i.e. the state-value $V(s)$, or state-action pairs, i.e. the action-value $Q(s, a)$. In deep RL $Q_\phi(s, a)$, also referred

to as the critic, is modelled by an ANN parameterized by $\phi$. Typically in actor-critic methods, the actor is updated such that $Q(s, a)$ is maximized:

$$\theta_{k+1} = \arg\max_{\theta} \mathbb{E}\left[Q_\phi(s, \pi_{\theta_k}(s))\right] \qquad (4)$$

For our application, the RL agent is a combination of the human and the ANN policy. The former contracts their muscles to execute an intended movement, and the latter maps the recorded EMG signals to the correct movement. Online RL presents unique challenges when humans are involved. Firstly, participants having to adapt to policy changes can lead to a less satisfactory experience due to inconsistency between policies. Secondly, the interaction between the human and the environment needs to be real-time, since significant lags between the commands of the human and the game would lead to actions $a_t$ in response to earlier states $s_{t-l}$, where $l$ is the lag. Applying online RL could slow down the game interface, creating such lags.

An alternative that has been gaining interest in the literature is offline RL (Levine et al., 2020), which aims to learn from a static dataset, $\mathcal{D}$. This solves the aforementioned problems but brings its own challenges. Because pure offline RL assumes no additional online data collection, it usually cannot reach acceptable online results without additional fine-tuning. The recent Advantage Weighted Actor-Critic (AWAC) algorithm proposed by Nair et al. (2020) aims to accelerate such online fine-tuning with offline datasets. AWAC trains an off-policy critic and an actor with an implicit policy constraint. This leads to the modified policy update as described as:

$$\theta_{k+1} = \arg\max_{\theta} \mathbb{E}\left[\log \pi_\theta \exp\left(\frac{1}{\lambda} A\left(s, \pi_{\theta_k}(s)\right)\right)\right] \quad (5)$$

where $A(s, a) = Q_\phi(s, a) - V(s)$, is the advantage function and $\lambda$ is the Lagrangian multiplier. By implicitly constraining the actor to stay close to the actions observed in the data, this algorithm was shown to be able both effectively train offline and to continue improving with experience, on real-world robotic problems.

# 5. Methods

We now present an overview of the proposed framework in Section 5.1, followed by the pretraining procedure in Section 5.2, the RL formulation of the problem in Section 5.3 and the evaluation procedure in Section 5.4.

## 5.1. Overview

Our framework is illustrated by the block diagram in Fig. 3. The starting point of our method is to execute the traditional procedure for training an SL policy. That is, we perform a recording session where the subject is prompted to repeatedly execute each movement for a certain duration of time. The resulting raw EMG data is then processed as described in Section 4.1 and labeled with the corresponding binary vector $m_i$ for each movement class $i$, to create an initial dataset, $\mathcal{D}_0 \sim (\mathcal{S}, \mathcal{A})$.

A policy $\pi_0$ is subsequently trained through SL using $\mathcal{D}_0$. This initial policy is then used to play a game similar to Guitar Hero. The goal of the game is to execute the correct movements, precisely initiating them at the right moment and sustaining them for the specified duration, as if to play notes in a song. After the participant finishes the first song, the new EMG data recorded during the game form a dataset $\mathcal{D}_1 \sim (\mathcal{S}, \mathcal{A}, \{-1, 0, 1\})$. This new dataset is then used to train a policy $\pi_1$ through an offline RL implementation, with policy $\pi_0$ as the starting point. The resulting policy replaces the previous one to play the song again. This process is repeated $n$ times, with each repetition appending the dataset i.e. $\mathcal{D}_1 \subset \mathcal{D}_2 \subset \ldots \subset \mathcal{D}_n$. For our experiments, we repeated the procedure for $n = 8$ times.

## 5.2. SL Pretraining

Each movement (see Fig. 1) is recorded 6 times, with a duration of 3 s for each repetition. The first and last 10% of each recording is discarded to omit transient EMG. Subsequently, the resulting $\mathcal{D}_0$ dataset is used to train an ANN described in Section 4.1. The second and fifth recordings are exclusively reserved as the validation set to determine the best model. Following 500 training epochs, the model with the highest F1 macro score on the validation set is selected as the baseline policy and will be referred to as the pretrained model, i.e. $\pi_0$.

Typically, multi-label classification problems treat each output as the probability of each class being present in the input, by minimizing the sum of binary cross-entropy losses of all classes. Instead, we use a simple RMSE loss, which is more generally used in Behavior Cloning applications (Azam et al., 2021) (i.e. SL for control applications).

## 5.3. RL Formulation

While in Section 4.2 we presented the general RL framework, here we introduce the details of our MDP formulation. We select $\gamma = 0.89$ and $\lambda = 0.95$, following a hyperparameter optimization process (see Appendix A). The state space $\mathcal{S} \in \mathbb{R}^{32}$ is defined as the stacked vector of 4 features for all 8 channels, as illustrated in Fig. 2. The action space $\mathcal{A} \in \{0, 1\}^7$ is the binary vector described in equation (1). Finally, the reward function is defined as:

$$r(s_t) = \begin{cases} 1 & \text{if} \quad a_t = a_t^* \land a_t^* \neq m_0 \\ 0, & \text{if} \quad a_t = a_t^* \land a_t^* = m_0 \\ -1, & \text{otherwise} \end{cases} \qquad (6)$$

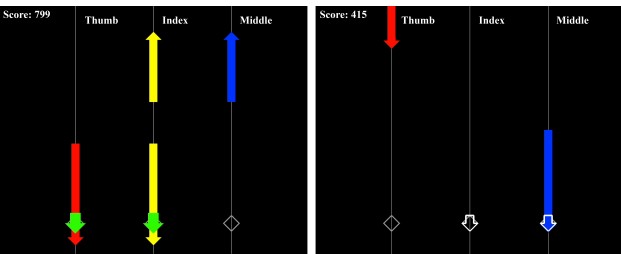

*Figure 4.* Game interface. Each vertical line refers to a controlled DOF: **Thumb** (red), **Index** (yellow) and **Middle** (blue). The arrows pointing up or down refer to **extension** and **flexion**, respectively. Desired movements are shown along the vertical lines, while predictions are displayed on the diamonds by short arrows, indicating the direction and DOF that is activated. When the agent executes the desired movement, a green arrow appears over the diamond of the specific DOF (left). Conversely, when the movement is incorrect the arrows are shown in white (right).

where $a_t^*$ is the correct movement at time $t$. The reward function plays a crucial role in the effectiveness of RL for any given task. Our objective is to train a policy that can accurately predict all movements made by the participant. As described by equation (6), we assign a reward of 1 to correct predictions and a reward of -1 to incorrect predictions. Furthermore, when no movement is desired. i.e. $a_t^* = m_0$, and the 'Rest' class is predicted we assign zero reward.

The final element of our RL formulation is the environment itself, represented by probability density function $p(s_{t+1}|s_t, a_t)$. To avoid ambiguous results and make it easier to evaluate the performance of the policy-human agent, we designed a task that involves distinct movements at defined times. This simplifies the process of assigning rewards to the policy. Moreover, in order to make the training process more engaging, we developed a serious game environment similar to Guitar Hero, shown in Fig. 4. Note that gamification has been shown to increase participant engagement in previous studies (Prahm et al., 2018).

The serious game (see Fig. 4) involves syncing movements to the beat of a song and displaying them in a way that mimics playing notes. The timing and correctness of the movements can be easily tested in this way. Each song corresponds to one RL episode, with movements lasting for 0.5, 1.0, 1.5, and 2.0 seconds. These lengths were chosen to align with the time needed for the average prosthetic hand to move, with 2 seconds being the maximum time elapsed from one extreme to the other. Each movement appears once for each length in one song, thus with 4 repetitions overall. Every repetition uses the same song to allow for a fair comparison. Each episode lasts for 137 seconds, with 60 seconds of these filled with notes. For this setting, each episode's undiscounted ($\gamma = 1$) return falls within the range $G_0 \in [-2740, 1200]$, which is used to normalize the re-

sults. However, seeing a negative score could demotivate the participants while interacting with the game. Therefore, we defined a scoring system that is always positive but follows the same trend as the episode's return. The score was displayed on the top left of the user interface to help participants keep track of their performance.

Based on the proposed RL formulation, we apply the AWAC algorithm introduced in Section 4.2 to train the myoelectric control policy. As one episode provides relatively little data, we include all recorded gameplay data in the replay buffer. In order to enhance the dataset and introduce exploration, we apply data augmentation by randomizing samples within dataset $\mathcal{D}_n$. To ensure a smoother playtime experience, the randomization process occurs after each episode, before training rather than in real-time. This randomization happens with a probability $\epsilon$ for samples with negative rewards. If triggered, a movement from Fig. 1 is selected based on a uniform probability distribution. Subsequently, a new reward is calculated for the selected movement.

### 5.4. Evaluation

Policy evaluation is important for two main purposes: (i) model selection and (ii) method validation.

**Model selection** is necessary because deep RL algorithms do not monotonically improve the learned policy. Indeed, determining if under- or over-fitting occurs is still an open problem (Kumar et al., 2021). Thus, in each training repetition, the best result may not be the last policy. Consequently, intermediate policies are tested based on all recorded game data so far. While the best procedure for model selection is to have the participant test each policy online, we want to minimize the amount of tests one must carry out, to prevent fatigue. Instead, during training, the song is simulated using the recorded data as input to each intermediate model to observe how the episode return improves. For each repetition, we choose the policy that has the highest simulated episode return, after 2000 gradient steps (see Appendix A). This showed increased performance compared to simply training for a predefined length and taking the last policy.

**Method validation** is a challenging task, as offline and online performances do not necessarily align in EMG control. Ideally, a classifier has to be tested online to come to a valid conclusion about its effectiveness. The episode return is such an online metric of performance, and we further compute the EMR and F1 macro scores, introduced in Section 4.1, for each song repetition. However, to not just evaluate our approach on gameplay data, which the pretrained policy did not train on, we also perform a separate online *Motion Test* (Kuiken et al., 2009). This test is performed at the end for both $\pi_0$ and $\pi_8$ in a randomized manner, to compare the SL policy with the final RL policy. During a *Motion Test*, participants are prompted to execute each movement a given

number of times, and the classification results are recorded. We have the participant repeat each movement 3 times in random order. For each trial, if the correct movement is predicted 40 times (i.e. for 2 seconds) the test is considered successful. Otherwise, a timeout of 10 seconds is reached. The next movement is prompted either after succeeding or after a timeout.

### 5.5. Experimental Setup

Since this study consists of a preliminary investigation of the potential of RL for prosthetic control, we first carry out our experiments on able-bodied individuals. We recruited nine participants to perform the experiments, with an age ranging from 23-28.

To familiarize the participants with the initial policy, $\pi_0$, a short version of the *Motion Test* was conducted (1 repetition and 5 seconds of timeout). Furthermore, to also get familiar with the game, one song was played for which the resulting data was discarded. After completing the proposed procedure, i.e. after the $n$-th iteration, the participants were asked to play the song one more time using the initial policy. This was important to be able to distinguish the impact of RL learning from just human learning. See our project page for visual examples sites.google.com/view/bionic-limb-rl.

The study protocols were carried out in accordance with the declaration of Helsinki. Signed informed consent was obtained from each participant before conducting the experiments. The study was approved by the Regional Ethical Review Board in Gothenburg (Dnr. 2022-06513-01).

## 6. Experimental Results

We found that the normalized average cumulative reward of the last RL repetition $\pi_8$ increased by 3 times (from $0.26 \pm 0.10$ to $0.78 \pm 0.13$) compared to the initial SL policy $\pi_0$ (see Fig. 5). The normalized average cumulative reward consistently increased from $\pi_1$ to $\pi_8$. While results appear to plateau, the observed increase between repetitions 7 and 8 implies that additional training could yield further benefits. Moreover, while the normalized average cumulative reward of the initial SL policy $\pi_0$ in the final repetition is higher than in repetition 0 ($0.28 \pm 0.13$ compared to $0.26 \pm 0.10$), it is still more than 2.5 times lower than the final RL policy $\pi_8$. This indicates that the RL approach had a substantial impact in improving performance, despite the human learning effect. These results were corroborated by the EMR more than doubling (from $0.30 \pm 0.16$ to $0.79 \pm 0.08$, see Fig. 6a) and a 40% increase in F1 scores (from $0.52 \pm 0.10$ to $0.75 \pm 0.15$, see Fig. 6b).

Substantial improvements were observed in most cases except for one participant (see the lower outliers depicted as circles in Fig. 5 and Fig. 6). This presents an opportunity to

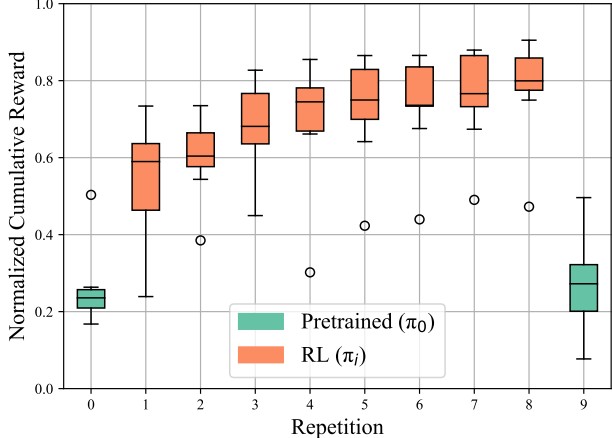

*Figure 5.* Normalized average cumulative reward over all subjects for RL training repetitions. The first and last repetition is done with the initial pretrained policy $\pi_0$, so RL training is only done between repetitions 0 and 8 using the most recent policy $\pi_i$. There are outliers in most repetitions, they belong to the same participant for whom RL did not seem to find patterns. The outlier in repetition 0 belongs to another person for whom pretraining worked exceptionally well.

further discuss the factors necessary for successfully using RL for decoding motor intent. Our hypothesis is that high variability in the participant's movements during gameplay, including incorrect movements, may have been the primary obstacle. It is noteworthy that some human error occurs for most participants during the training process. While this may prolong the learning time, it does not seem to hinder improvement for most participants. This finding highlights the interdependence between human and policy. Nevertheless, we believe that with continued training of both entities, RL could eventually also exhibit improvements for this participant, but further experiments are necessary.

An additional comparison was performed based on the outcomes of the *Motion Test* in order to understand if the improvements described above were still observed in a context different from the game environment. Indeed, the mean EMR of the RL policy $\pi_8$ during the *Motion Test* doubled (from $0.23 \pm 0.14$ to $0.47 \pm 0.19$) compared to the pretrained policy $\pi_0$ (see Fig. 6c). Moreover, notable improvements were observed in the F1 macro score, with an increase of over 15% (from $0.58 \pm 0.11$ to $0.68 \pm 0.19$) (see Fig. 6d).

One possible reason for the lower observed metrics in the *Motion Test* compared to the game environment could be the difference in the participant's focus between the two settings. During gameplay, the primary focus is on the game, thus mainly on the correct timing of movements, while in the motion test, the participant's sole focus is on how to execute the movement. This shift in attention may result in altered muscle activations, potentially contributing to the variations in observed metrics.

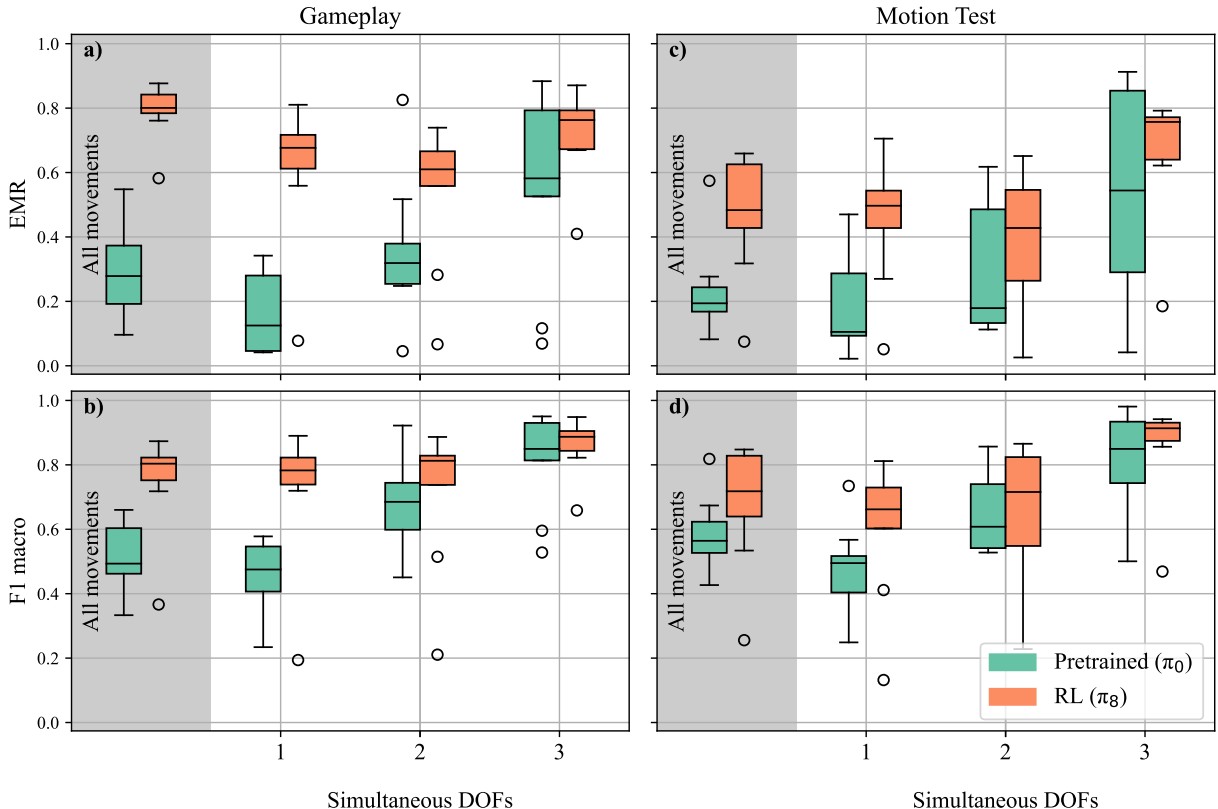

*Figure 6.* EMR and F1 macro for all movements in Fig. 1 and per DOF for both Gameplay and Motion Test. Note that all movements also include $m_0$, 'Rest', which is not included in the box plots per DOF. For Gameplay, repetition 8 with policy $\pi_8$ and repetition 9 with $\pi_0$ are compared. Every measure in all scenarios improves with RL, especially for movements that performed poorly before, namely 1 DOF classes. Similar to before, the lower outliers in RL belong to one subject where RL did not improve performance. Detailed numerical values can be found in Table 2 in the Appendix.

We further noticed that single DOF movements demonstrated substantial improvement, while movements involving 2 or 3 simultaneous DOFs displayed less marked changes (see Fig. 6a-d). Single DOF movements might be more challenging to decode due to their relatively lower muscle activation compared to movements with multiple DOFs, explaining the lower EMR and F1 score for single DOF movements. On one hand, lower scores inherently provide more room for improvement. On the other hand, this could also mean that the RL approach primarily focuses on refining movements that initially perform poorly. These trends are consistent across both the game environment and the *Motion Test*, with slightly more pronounced effects observed in the game environment.

## 7. Conclusion

In this study, we leveraged RL to improve the decoding of motion intent with the aim of creating more intuitive and responsive bionic limb controllers. The proposed method

using an interactive game substantially enhanced the online, human-in-the-loop performance of an ML controller, thus bridging the gap between offline training and real-life usage.

To validate the general efficacy of our method, the same experiment will be repeated with people with amputation in future work. Since our approach is rather general, it can be extended to other serious games more representative of daily prosthetic use. Notably, our work could also be extended to allow fine-tuning based on human feedback and thereby improve prosthetic functionality to better accomplish tasks encountered in daily life.

## Acknowledgments

This project was funded by the Promobilia Foundation, the IngaBritt and Arne Lundbergs Foundation and the Swedish Research Council (Vetenskapsrådet). Additionally, this work was partially supported by the Wallenberg AI, Autonomous Systems and Software Program (WASP) funded by the Knut and Alice Wallenberg Foundation.

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

# A. Implementation Details

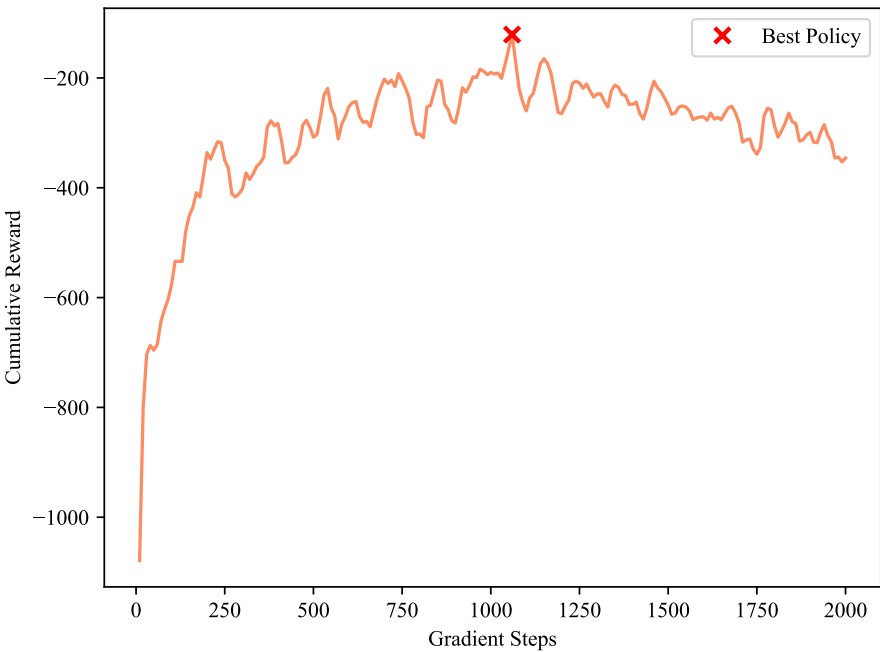

*Figure 7.* Progression of simulated reward during training. The policy is evaluated every 10 gradient steps.

To determine the best policy during training we employ a simulation of playing a song and calculate the cumulative reward at intervals of 10 gradient steps. After training for 2000 gradient steps, we select the policy with the highest reward as the preferred choice for further progression. In Fig. 7, we present a typical reward progression during the early repetitions. Initially, the reward increases, but it gradually begins to decline. Hence, selecting the last policy would be detrimental. Unlike in SL, where training is often stopped when the loss fails to decrease after a certain number of epochs, we avoid such early stopping. RL training can exhibit instability, and the simulated reward may continue to increase even after several decreasing gradient steps. Thus, we opt for a fixed number of 2000 gradient steps to ensure sufficient training.

In addition to early stopping, we performed an automated parameter search to optimize our hyperparameters. During this process, the best policy is chosen as described above using training data and evaluated on test data that it did not see during training. Using training data for choosing the best model was necessary because, during online training, we lack access to a test song that could solely serve this purpose. The hyperparameters yielding the highest test reward are summarized in Table 1 and are employed for RL training in our experiments.

*Table 1.* Hyperparameters used for RL training. The values are found by a hyperparameter search where the policy with highest test reward was chosen.

| HYPER-PARAMETER | VALUE |
| --- | --- |
| RL BATCH SIZE | 512 |
| DISCOUNT FACTOR $\gamma$ | 0.8935 |
| WEIGHT CALCULATION $\lambda$ | 0.95 |
| REWARD SCALING | 1 |
| POLICY WEIGHT DECAY | $10^{-4}$ |
| POLICY LEARNING RATE | $9.844 * 10^{-4}$ |
| Q HIDDEN SIZES | $[256, 256]$ |
| Q HIDDEN ACTIVATIONS | RELU |
| Q WEIGHT DECAY | 0 |
| Q-FUNCTION LEARNING RATE | $7.627 * 10^{-4}$ |
| NUMBER OF CRITICS | 2 |
| TARGET NETWORK SYNCHRONIZATION COEFFICIENCY $\tau$ | $8.948 * 10^{-3}$ |
| NUMBER OF ACTION SAMPLED TO CALCULATE $A^{\pi}(s_t, a_t)$ | 1 |
| N-STEP TD CALCULATION | 1 |
| Q-FUNCTION | MEAN APPROXIMATOR |
| INTERVAL TO UPDATE POLICY | 4 |
| CHANCE TO RANDOMIZE WRONG NOTES $\epsilon$ | 0.9 |

*Table 2.* Comparison of classification results during gameplay and *Motion Test*, using policy $\pi_8$ (in episode 8), the initial policy, $\pi_0$ (in episode 9). Results are described by the EMR, F1 and their standard deviation, across all participants.

| | GAMEPLAY | | MONTION TEST | |
| --- | --- | --- | --- | --- |
| | EMR | F1 MACRO | EMR | F1 MACRO |
| $\pi_0$ | $0.30 \pm 0.16$ | $0.52 \pm 0.10$ | $0.23 \pm 0.14$ | $0.58 \pm 0.11$ |
| $\pi_8$ | $\mathbf{0.79} \pm 0.08$ | $\mathbf{0.75} \pm 0.15$ | $\mathbf{0.47} \pm 0.19$ | $\mathbf{0.68} \pm 0.19$ |