# OpenReview forum: "Improving Bionic Limb Control through Reinforcement Learning in an Interactive Game Environment"
_ICML.cc/2023/Workshop/ILHF — ILHF Workshop ICML 2023_

### Official Review · Reviewer_xiMn · 2023-06-04
**Reasonable for the workshop**

**Rating:** 6
**Confidence:** 3

**Review:**

This paper studies the interesting application of bionic limb control, which involves the design of prostheses that can assist humans who have limb amputations. An important factor in the usage of such prosthetics is whether the user’s motion intent can be decoded to determine the appropriate bionic joint actuation. It is possible to use machine learning for this but for supervised learning it is hard to get labeled data; basically, the human must move the prosthetics, and then someone has to record "electromyograhic" data which is hard. This is why the paper proposes to use reinforcement learning in an interactive game environment. They develop an environment which is like the "Guitar Hero" game.

The paper seems to be reasonably good for a workshop. While I am not an expert in this specific area it seems reasonable. A few questions and suggestions:

Question: why say "batch" reinforcement learning in the title? I think there is some online interaction here so it might be simpler to remove that word.

Question: how much "implicit" feedback do we get from the RL process? The workshop is on "implicit" human feedback but the feedback seems to be very explicit here.

Suggestion: restate the project website link in the abstract and introduction. It helps the reader to see the bionic limb control in action.

Suggestion: I would encourage slightly more anonymization since the study was approved by a board in Gothenburg (a city in Sweden). However I would not desk-reject the paper since everything else is well anonymized.

Suggestion: I would merge Section 3 (Motivation) into the introduction and related work, otherwise the motivation seems too spread out thin throughout the paper.

---

### Official Review · Reviewer_CKDC · 2023-06-15
**Exciting application**

**Rating:** 7
**Confidence:** 3

**Review:**

The paper demonstrates success in using reinforcement learning to further train a model predicting user intended motion from biological signals for users of bionic limbs.

Pros:
1. The motivation is really strong: the lack of good supervised data in this domain is very clear, so it is exciting that an RL-based approach can be successful
2. The writing is clear even for readers unfamiliar with the domain

Cons:
1. It is not clear how limited the proposed approach is to specific RL environment it is trained in, and therefore its generality
2. Does the approach work in environments where there is no clear reward / non-game environments? It is unclear if the problem of reward specification is strictly easier than gathering supervised data

---

### Decision · Program_Chairs · 2023-06-20

Accept